# Differential Modulation of Dendritic Cell Biology by Endogenous and Exogenous Aryl Hydrocarbon Receptor Ligands

**DOI:** 10.3390/ijms24097801

**Published:** 2023-04-25

**Authors:** Atefeh Sadeghi Shermeh, Dmytro Royzman, Christine Kuhnt, Christina Draßner, Lena Stich, Alexander Steinkasserer, Ilka Knippertz, Andreas B. Wild

**Affiliations:** Department of Immune Modulation, Universitätsklinikum Erlangen, Friedrich-Alexander-Universität Erlangen-Nürnberg, 91052 Erlangen, Germany; dmytro.royzman@uk-erlangen.de (D.R.); christine.kuhnt@uk-erlangen.de (C.K.); christina.drassner@uk-erlangen.de (C.D.); lena.stich@uk-erlangen.de (L.S.); alexander.steinkasserer@uk-erlangen.de (A.S.); ilka.knippertz@uk-erlangen.de (I.K.)

**Keywords:** aryl hydrocarbon receptor, dendritic cells, exogenous ligands, endogenous ligands

## Abstract

The aryl hydrocarbon receptor (AhR) is a decisive regulatory ligand-dependent transcription factor. It binds highly diverse ligands, which can be categorized as either endogenous or exogenous. Ligand binding activates AhR, which can adjust inflammatory responses by modulating immune cells such as dendritic cells (DCs). However, how different AhR ligand classes impact the phenotype and function of human monocyte-derived DCs (hMoDCs) has not been extensively studied in a comparative manner. We, therefore, tested the effect of the representative compounds Benzo(a)pyrene (BP), 6-formylindolo[3,2-b]carbazole (FICZ), and Indoxyl 3-sulfate (I3S) on DC biology. Thereby, we reveal that BP significantly induces a tolerogenic response in lipopolysaccharide-matured DCs, which is not apparent to the same extent when using FICZ or I3S. While all three ligand classes activate AhR-dependent pathways, BP especially induces the expression of negative immune regulators, and subsequently strongly subverts the T cell stimulatory capacity of DCs. Using the CRISPR/Cas9 strategy we also prove that the regulatory effect of BP is strictly AhR-dependent. These findings imply that AhR ligands contribute differently to DC responses and incite further studies to uncover the mechanisms and molecules which are involved in the induction of different phenotypes and functions in DCs upon AhR activation.

## 1. Introduction

The aryl hydrocarbon receptor (AhR) was initially identified as a cytosolic sensor for the potent environmental pollutant 2,3,7,8-tetracholrodibenzo-p-dioxin (TCDD) [1]. However, increasing evidence has shown that the ligand–AhR complex functions as a transcriptional activator, which mediates various biological processes, including immune regulation [2,3]. Activation of the AhR leads to multifaceted cellular responses, owing to (i) its vast range of target genes, (ii) the nature of the ligand, and (iii) the environmental context of the targeted cell [4]. AhR ligands can be categorized according to their origin into exogenous (natural and synthetic) and endogenous (host- and microbiota-derived) agonists [5,6,7]. Upon ligand binding, AhR is subjected to conformational changes, which activate a cascade of interactions including AhR–ligand translocation to the nucleus and AhR binding to the AhR nuclear translocator (ARNT), which ultimately leads to the expression of target genes harboring xenobiotic response elements (XREs) [3]. Amongst others, AhR target genes comprise cytochrome P450 superfamily enzymes (*CYP1A1*/*2*, *CYP1B1*), which aid in ligand metabolization [8,9,10]. Together with the AHR repressor (*AHRR*), whose transcription is also induced by AhR, these gene transcripts provide a negative feedback loop to control AhR activity. In addition to the genomic (canonical) pathway, AhR activation controls cellular responses to its ligands via a non-genomic route, i.e., via the proteasomal degradation of other transcription factors, and by the regulation of kinase-signaling events [3,8].

High expression of AhR has been reported in cells of the immune system including Th17, regulatory T (Treg) cells, and dendritic cells (DCs) [3,11]. DCs are skilled antigen-presenting cells (APCs) that orchestrate the differentiation, activation, and polarization of T cells [11,12,13]. Due to their localization in the body and their antigen-presenting role, DCs are constantly exposed to AhR agonists. Depending on the nature of the ligand, AhR activation can induce either immunogenic or tolerogenic phenotypes in DCs, which lead to an extensive impact on T cell immunity [3,14,15].

Benzo(a)pyrene (BP), a synthetic exogenous ligand, belongs to the polycyclic aromatic hydrocarbons (PAH) with potent mutagenic and carcinogenic properties. BP metabolization is mediated by several CYP450 family members after its binding to AhR [16,17]. In addition to its carcinogenic effects, BP is considered an immunotoxic agent, which is capable of inhibiting B and T cell proliferation, cytotoxic T cell induction, and antibody production, as shown in the murine system [18,19,20]. Human and murine studies also indicated impaired differentiation and maturation in both bone marrow-derived DCs (BMDCs) as well as blood monocyte-derived DCs (MoDCs) treated with BP [21,22], although the cellular mechanisms are not fully understood.

Unlike BP, the endogenous agonist 6-formylindolo[3,2-b]carbazole (FICZ) is a photoproduct of the essential amino acid tryptophan (Trp) with no described cytotoxicity during normal intake [23]. Despite its physiological origin and rapid degradation through the AhR pathway, FICZ administration boosts Th17 cell responses and exacerbates autoimmunity [24,25]. Most of the human and murine research emphasizes on the activation of the Trp-catabolizing enzyme indoleamine 2,3-dioxidase (IDO), following the exposure of DCs to FICZ [26,27,28]. However, this effect seems to be ligand-unspecific, since other AhR ligands including TCDD, kynurenine, and 2-(10H-indole-30-carbonyl)-thiazole-4-carboxylic acid methyl ester (ITE) show the same effect [23,29,30]. Currently, attention is drawn to another important group of tryptophan-derived AhR ligands, i.e., endogenous indoles. Indoxyl 3-sulfate (I3S) is an indole metabolite that is produced when microbiota-derived tryptophanase converts dietary tryptophan to indole, which is transported to the liver and metabolized by sulfotransferase [31]. A murine study (in vitro and in vivo) indicated that I3S activation of AhR stimulates Th17 differentiation and exacerbates experimental autoimmune encephalomyelitis (EAE) [32]. Interestingly, another study reported that I3S-treated MoDCs show an anti-inflammatory phenotype due to the suppression of the NF-kB pathway [33]. Altogether, AhR activation affects the phenotype of immune cells, and DCs in particular, in a highly ligand-dependent manner. However, how diverse ligands induce specific biologic outcomes and what molecular targets and mechanisms are involved in the induction of immunogenic or tolerogenic phenotypes in human DCs remain not fully understood.

Thus, in the present study, we analyze the effect of different AhR–ligand classes on DC biology. We chose BP as a representative for exogenous ligands and FICZ for host endogenous ligands. I3S, which originates from tryptophan becoming catalyzed by gut bacteria enzyme that is then modified by endogenous enzymes, represents ligands of dual origin (microbiota and host). Thereby, we disclosed that although all three ligands activated the AhR pathway and reduced the expression of the immunoregulatory molecule CD83, only BP constantly increased negative regulators including members of the immunoglobulin-like transcript (ILT) family. Moreover, at a functional level, BP-treated DCs are restricted in their ability to induce T cell proliferation by shifting the ratio of Treg to effector T cell in DC–T cell co-cultures. Using CRISPR/Cas9-mediated knock-down (KD) of *AHR* in DCs, we further demonstrated that the impact on DC phenotype and the stimulatory capacity of DCs is clearly AhR-dependent. Collectively, our data show that the nature of AhR ligands directly affects DC function, which greatly adds to understanding the role of the AhR-pathway in immunity.

## 2. Results

### 2.1. AhR Activation in DCs Results in Phenotypic Changes, Which Depend on the Class of AhR Ligand

Since a broad comparison of exogenous, endogenous, and dual-origin ligands has not yet been performed in human DCs, we first investigated the capacity of BP, FICZ, and I3S to activate the AhR pathway in these cells. To this end, cells were pre-treated with 1 µM BP, 100 nM FICZ, 500 µM I3S, and DMSO as a vehicle control (Mock) for 6 h, and then matured in the presence of LPS for an additional 16 h. To assess the activation of the AhR pathway, we analyzed the expression of known target genes such as *CYP1A1*, *CYP1B1*, and *AHRR*. Although all tested ligands induced the expression of *CYP1A1*, *CYP1B1*, and *AHRR*, we detected differences in the intensity of activation. BP and FICZ were equally capable of inducing *CYP1A1* and *CYP1B1* transcripts, while we observed less up-regulation after treatment with I3S (Figure 1A–C). By contrast, both I3S and FICZ showed strong induction of *AHRR*, while BP was less strongly induced this transcript (Figure 1D). Interestingly, BP significantly reduced mRNA expression of *AHR* itself (Figure 1E), an effect which was further corroborated by Western blot analyses showing lower protein levels of AhR only in BP-treated cells (Figure 1F).

In previous studies, we revealed that AhR activation by the flavonoid quercetin results in a reduction in CD83 expression, a prominent marker of mature DCs [15]. When we analyzed the expression of CD83 in our ligand-treated DCs, we observed a strong suppression of LPS-mediated CD83 induction by all the used ligands, both on the mRNA and protein level (Figure 1G,J, respectively). Interestingly, while HLA-DR and CD86 were not influenced by any ligand (Appendix A, respectively), the co-stimulatory molecule CD80 was strongly repressed upon BP treatment, but not by FICZ or I3S (Figure 1H,K). Contrarily, we detected significantly increased transcripts of *IL2RA* (*CD25*) after AhR activation, regardless of the ligand (Figure 1I), while only FICZ resulted in an elevated surface expression of the corresponding protein (Figure 1L). These data provide evidence that different classes of AhR ligands vary in their activation of AhR target genes and subsequent effect on the phenotype of DCs.

### 2.2. Induction of a Tolerogenic DC-Phenotype Is Ligand-Dependent

Since we observed a dysregulation of classical DC activation markers upon treatment with different AhR ligands, we reasoned that these might induce a tolerogenic state in DCs. Such tolerogenic DCs (tolDCs) are characterized by the expression of inhibitory molecules that interfere with immunogenic activation. Amongst others, tolDCs express a group of immunoglobulin-like transcripts (ILTs) that contain immunoreceptor tyrosine-based inhibitory motifs (ITIMs) in their cytoplasmic domain [34]. When we assessed the expression of major inhibitory ILTs, the pre-treatment of DCs with BP strikingly increased the level of *ILT3*, *ILT4*, and *ILT5* transcripts in the DCs (Figure 2A–C), whereas FICZ and I3S failed to induce *ILT3* and *ILT4*. Only *ILT5* expression was significantly promoted by I3S treatment (Figure 2C). Similarly, when we analyzed the surface expression of the corresponding proteins by flow cytometry, we detected a significant induction of all three ILTs in the BP-treated cells (Figure 2D–F). Interestingly, I3S, while having no impact on ILT3 or ILT5 expression, significantly reduced ILT4 surface levels (Figure 2E). In contrast to mRNA data, FICZ significantly induced ILT5 expression (Figure 2F). Thus, although all tested ligands were able to activate AhR-dependent pathways, only BP consistently induced the expression of inhibitory ILTs in the DCs.

In addition to ILTs, a further characteristic of tolDCs is the expression of ectonucleotidases such as CD39 and CD73, which catalyze the degradation of pro-inflammatory extracellular ATP (eATP) into anti-inflammatory adenosine [35]. When we investigated mRNA levels of both *CD39* and *CD73*, we revealed an upregulation of *CD39* after treatment with FICZ and I3S (Figure 2G), while *CD73* expression levels were not altered, when compared to the mock control (Figure 2H). In contrast to mRNA results, both endogenous and exogenous ligands upregulated CD39 expression in protein levels (Figure 2I). Interestingly, flow cytometric analyses identified increased protein levels of CD73 in BP-treated cells, but not in endogenous ligand-treated cells (Figure 2J). Since CD39 and CD73 are responsible for degrading eATP, we also assessed the metabolization of this pro-inflammatory mediator by ligand-treated DCs. In line with mRNA data regarding CD39 expression, FICZ-treated DCs significantly catabolized more ATP compared to mock-treated cells, and a trend was observed for the BP-treated DCs (Figure 2K).

### 2.3. BP Treatment Impairs the Migratory Capacity of Mature Human DCs

One important feature of mature DCs is their capacity to migrate following a chemotactic gradient, which directs them to lymph nodes where they then activate antigen-specific T cells. Therefore, mature DCs express high levels of the chemokine receptor CCR7. When we assessed the impact of AhR activation on CCR7 expression, BP-treated cells showed a highly significant reduction in *CCR7* mRNA levels (Figure 3A), while FICZ and I3S had no effect. In terms of protein levels, BP induced a slight reduction, while FICZ or I3S pre-treated DCs showed a significant upregulation of CCR7 expression levels (Figure 3B). We also evaluated the functional relevance of these changes by assessing the migratory capacity of AhR-activated DCs towards the chemokine CCL19, which is the ligand for CCR7. Here, we observed the significantly suppressed migratory capacity of BP-treated DCs (Figure 3C), while FICZ exposure induced increased migration potential, though this difference was not significant (Figure 3C).

### 2.4. BP Treatment Induces a Regulatory DC Phenotype Which Correlates with Reduced T Cell Stimulatory Capacity

Since we observed the induction of tolDC-associated markers after the treatment with specific AhR ligands, we wondered if this would result in insufficient T cell stimulation by these cells. To assess the ability of AhR ligand-treated DCs to induce T cell proliferation, we co-cultured AhR ligand-pre-treated and then LPS-matured DCs together with allogeneic responder cells and measured their proliferation by BrdU incorporation. We observed a significantly reduced proliferation of T cells that were co-cultured with BP-pre-treated DCs (Figure 4A). Interestingly, this was not the case when endogenous ligands were used. In the next step, we further aimed to interrogate the effect of AhR-mediated DC activation regarding its influence on T cell differentiation. To this end, we isolated T cells from PBMCs by magnetic separation and co-cultured them with AhR ligand-treated, LPS-matured DCs. Noteworthily, when the DCs were pre-treated with BP, a significantly increased ratio of Treg (CD25^high^FoxP3^+^) to effector T cells (CD45RO^+^CD69^+^) was observed (Figure 4B). Again, endogenous ligands showed no differences compared to the mock condition. This suggests that pre-treatment with BP induces tolDCs, which shift the ratio of T effector cells towards an immunosuppressive Treg phenotype.

### 2.5. Efficient CRISPR/Cas9-Mediated AHR Knock-Down in Human DCs

To exclude that the described alterations in DC phenotype and function are caused by side effects of the used ligands, we employed a CRISPR/Cas9-based strategy to delete *AHR* expression in DCs. We used a dual sgRNA approach to assure proper disruption of the *AHR* gene within exon 2, while control cells were only electroporated with Cas9 alone to rule out side effects related to electroporation or Cas9 itself. To check the efficiency of the CRISPR/Cas9 editing, genomic regions neighboring the cut site were amplified and sent for sequencing. The edited and control sequences, surrounding the guide sequences, are shown in Appendix A. Data analyses revealed a 100% indel (Appendix A), which indicates the percentage of CRISPR-edited sequences, resulting in a putative knock-down. Additionally, qPCR analysis on *AHR* mRNA level revealed a highly significant reduction in *AHR* expression in specific sgRNA-treated cells, in comparison to control cells (Figure 5A–D). To confirm the efficiency on protein level, AhR-specific Western blot analyses were performed, further confirming the *AHR*-specific knock-down (Figure 5E). Following exposure to the different AhR ligands, the level of AhR-induced transcripts, i.e., *AHRR*, *CYP1A1*, and *CYP1B1* was increased in ligand-treated control cells (Figure 5F–Q, respectively). In sharp contrast, adding ligands to *AHR*-KD cells did not cause induction of AhR-induced transcript, when compared to mock-treated cells.

### 2.6. Deletion of AHR in DCs Reverses the Tolerogenic Effects of BP

To evaluate the functional effect of our deletion strategy, both control and *AHR*-KD cells were treated with BP, FICZ, I3S, or DMSO for 6 h and were then matured by adding LPS for 16 extra hours, followed by gene expression analyses. Strikingly, the deletion of the *AHR* gene reversed the BP-mediated repression of *CD83* (Figure 6A–D). Likewise, BP failed to induce the expression of *ILT3* and *ILT5* in *AHR*-KD cells, demonstrating that the modulating effect of BP on DCs is clearly AhR-dependent. (Figure 6E–H,I–L, respectively). By contrast, *ILT4*, *CD39*, and *CD73* expression was not significantly altered between the control and *AHR*-KD DCs (Appendix A). Next, we assessed whether or not *AHR* deletion restores the diminished T cell stimulatory capacity of BP-treated DCs. Allogeneic T cells proliferated significantly less when co-cultured together with BP-treated wild-type control cells (Figure 6M), which resembled the results obtained from non-electroporated cells (Figure 4A). In sharp contrast, the suppressive effect was completely abolished, when BP-treated *AHR*-KD cells were co-cultured with allogeneic responder cells (Figure 6N). This further supports the fact, that BP acts via AhR. Again, we did not observe any differences in T cell stimulation, when *AHR*-KD DCs were pre-treated with FICZ or I3S, compared to mock-treated cells.

We provided evidence that different classes of AhR ligands significantly vary in their impact on DC biology and consequently, in affecting the immune response (summarized in Figure 7). Thereby, the exogenous ligand BP exhibited the strongest influence by enhancing tolerogenic features in DCs, ultimately leading to impaired T cell stimulation. This provides crucial insights into the differential regulation of the AhR pathway in immune cells.

## 3. Discussion

Categorizing AhR ligands based on their origin, structure, affinity to AhR, and metabolism rate (duration of presence in the body) allows a better understanding of the potency and efficacy of their interaction with AhR, as well as the functional activity of AhR as a transcription factor. Comprehending the potential of ligands with selective AhR modulator (SAhRM)-like activities provides insight into the therapeutic applications of AhR. This is particularly relevant in the context of dendritic cells (DCs), which express AhR and can be affected by its effects on their phenotype and function, especially in the context of DC–T cell interactions [35]. Nevertheless, only little is known about the interplay between AhR, AhRR, and the CYP450 family of enzymes in DCs. The present study revealed that treating hMoDCs with different AhR ligands (i.e., BP, FICZ, and I3S) could activate the AhR genomic pathway independent of ligand class. However, the extent of induced expression of three selected AhR-targeted transcripts (*CYP1A1*, *CYP1B1*, and *AHRR*) was different. Although the induction of *CYP1A1* following BP and FICZ treatment in hMoDCs has been reported before [21,36], to the best of our knowledge, this is the first time that expression of AhR-induced transcripts by I3S treatment has been reported in hMoDCs. We also observed an inverse expression pattern for *CYP1A1* and *AHRR* in the BP-treated DCs, compared to the DCs with I3S treatment. These data are in line with reports on the tissue- and cell-type-dependent reciprocal activities of *AHRR* and *CYP1A1* [37]. One explanation for this ligand-specific pattern might be that the cell tries to dispose of xenobiotic BP and thus, AhR activation after BP treatment leads to less inhibition via AhRR and a subsequently increased expression of metabolizing enzymes, such as CYP1A1. Conversely, exposure of DCs (unlike hepatocytes) to tryptophan metabolites might restrict AhR activation by the high expression of *AHRR*. Interestingly, there is still a significant level of expression of *CYP1A1* in DCs treated with FICZ, despite a high induction of *AHRR*. This is likely because FICZ has the highest affinity to AhR among endogenous ligands, and is also the most prominent substrate for CYP1A1 among this group of ligands [23]. In contrast to CYP1A1, whose expression is reportedly rather tissue-imprinted, CYP1B1 is more consistently expressed in different tissues, and its level increases due to the exposure of AhR ligands, especially PAHs [22]. Consequently, we detected increased *CYP1B1* levels in all three ligand treatments. Surprisingly, we observed that BP treatment remarkably reduced AhR expression both on mRNA and protein levels. As BP exhibits no overt toxic effects on hMoDCs (data are not shown) [21], this reduction is not caused by extensive cell death. However, it might be possible that BP-derived metabolites suppress AhR expression in a regulatory feedback loop to prevent the over-activation of this pathway.

DCs play a crucial role as gatekeepers and regulators of the immune system, and activation of AhR impacts their function [35]. Here, we show that AhR ligands of any class caused a significant decrease in CD83 expression, which is a prominent marker of mature DCs, while the co-stimulatory molecule CD80 was only affected by treatment with BP (both mRNA and protein levels). This is consistent with a previous report by Laupeze et al. who also observed a decrease in CD83 protein expression due to BP exposure in hMoDCs [21]. Concerning the impact of endogenous ligands on the expression of specific DC markers, the reported data are inconsistent, as two studies demonstrated a significant reduction in CD83 surface expression due to FICZ treatment [36,38], while another study indicated a significant suppression of CD80, CD86, and HLA-DR, but not of CD83 [39]. The reduction in CD83 expression observed in DCs treated with I3S contrasts the findings of Ghimire et al., who reported significant inhibition only for CD80 and CD86 protein expression levels [33]. The different outcomes could be explained by factors such as the experimental conditions (ligand concentration and exposure time) as well as donor-dependent confounders such as gender, age, immune system status, and metabolic rate.

In contrast to the observed repression of CD83, we revealed a noteworthy increase in *CD25* mRNA expression by any of the three ligands, which might relate to the role of CD25 in the induction of a regulatory DC phenotype. In this regard, heightened levels of CD25 may equip DCs to scavenge more IL2 and thus compete with T cells for this molecule, thereby facilitating the establishment of a tolerogenic environment [40,41]. The increased expression of CD25 might thus indicate the transformation of DCs into tolDCs, which can be further characterized based on their expression of specific inhibitory molecules [35]. Research has demonstrated that increasing the levels of ILT3 and ILT4 in monocytes and DCs can facilitate these antigen-presenting cells to effectively deactivate CD4^+^ Th cell responses [42]. On the other hand, AhR activation has already been reported to induce tolDCs by enhancing the conversion of eATP to anti-inflammatory adenosine by the ectoenzyme CD39 [43]. In this context, our data revealed that BP-treated DCs exhibited the highest upregulation of negative regulatory molecules, followed by those treated with FICZ. This upregulation can lead to the activation of transcriptional programs associated with tolDCs. These tolDCs have been identified by their ability to impair T cell proliferation and facilitate the development of regulatory T cells. Our findings demonstrated that BP-treated DCs inhibited T cell proliferation and induced Treg differentiation, which is consistent with the findings of a previous study [21]. However, treatment with endogenous ligands (FICZ and I3S) did not affect T cell proliferation or Treg cell induction, contradicting previous reports on their ability to induce regulatory T cells or inhibit T cell proliferation [33,38]. Importantly, BP-treated DCs also expressed significantly higher levels of CD73, which converts AMP into adenosine, a highly immune-suppressive agent involved in Treg induction [44]. Therefore, BP-induced tolerogenic programs in DCs lead to a modulated stimulatory capacity of these cells and consequently, to an impaired initiation of immune responses.

Furthermore, mature DCs exhibit high CCR7 expression levels, enabling them to migrate via the lymphatics following the chemoattractants CCL19 and CCL21, which direct them to the regional lymph nodes to present their captured antigens to specific T cells [45]. Earlier studies reported a decrease in the migration of murine BMDCs (both in vivo and in vitro) following AhR ligand treatment [46,47,48]. In addition, a downregulated migratory capacity of human MoDCs was also reported after the treatment with quercetin [15]. Our findings are consistent with those of these studies showing that the reduction in CCR7 expression levels interferes with the migration of these cells. Taken together, BP treatment not only affects the acquisition of an immunogenic phenotype but also impairs the migratory capacity of DCs, thereby severely interfering with their stimulatory function during immune responses.

Previous research suggested that the inhibitory phenotype induced in DCs by AhR ligands, particularly BP [21], may not be directly linked to the AhR, but rather contributes to the immunosuppressive effects of these ligands, especially PAHs. Therefore, we used a CRISPR/Cas9 approach to silence the *AHR* gene in hMoDCs and examined their phenotypic and functional changes post-exposure to AhR ligands. Our findings showed that if *AHR* is silenced, the induction of AhR-related transcripts following treatment with ligands is prevented. Furthermore, the effect of AhR ligands, particularly BP, on the DC marker *CD83* and the induction of inhibitory regulators *ILT3* and *5*, is abolished in the absence of AhR. Moreover, deleting the *AHR* gene reversed the antiproliferative effect of BP on DCs at the functional level. Thereby, we also rule out the possibility that reduced T cell proliferation is caused by a “spill-over” of AhR ligands from the DC to the T cell.

In conclusion, our data demonstrate that AhR activation, following exposure to the three ligands used in this study, modulates the DC phenotype. However, only BP consistently and effectively induced a phenotypic and functional modulation which was associated with a tolerogenic outcome. Although previous studies [49,50] suggest that natural AhR ligands, both exogenous and endogenous, can have SAhRMs-like activities and induce a tolerogenic phenotype in DCs, our data indicate that even a xenobiotic ligand such as BP can induce regulatory effects, which, however, may depend on tissue-, organ-, or species-specific factors. In this regard, BP might hijack the AhR pathway to impair DC-elicited tumor immunosurveillance, which contributes to its carcinogenic potential. By contrast, AhR-induced tolDCs could be considered powerful tools for immunotherapy, particularly in the context of autoimmune diseases [35]. It is important to note that all effects were observed in LPS-matured DCs, which rely on NF-kB signaling. As AhR is known to interact with this pathway, maturation with other agents (such as polyinosinic:polycytidylic acid (poly(I:C), which mainly acts via the IRF transcription factor) might yield different results [36]. Therefore, additional research on the function of AhR ligands and their metabolites as well as AhR signaling is necessary to enhance our comprehension of the mechanism by which AhR induces tolerogenicity in DCs.

## 4. Materials and Methods

### 4.1. Generation of Human Monocyte-Derived Dendritic Cells

Leukoreduction system chambers (LRSCs) from healthy volunteers (local ethics vote reference number: 357_19B) were used to generate human monocyte-derived immature DCs (iDCs) as previously stated [51]. Concisely, whole peripheral blood mononuclear cells (PBMCs) were isolated by density gradient centrifugation using Lymphoprep (Axis-Shield/Alere Technologies AS, Oslo, Norway). Blood monocytes were isolated from PBMCs by plastic adherence. Thus, PBMCs were plated in tissue culture dishes (Falcon, Fisher Scientific, Waltham, MA, USA) in a complete DC medium, which was composed of RPMI 1640 (Lonza, Gampel, Switzerland) and enriched with 10 mM HEPES (Lonza), 1% (*v*/*v*) penicillin/streptomycin/L-glutamine (Sigma-Aldrich, St. Louis, MO, USA), and 1% (*v*/*v*) heat-inactivated human serum type AB (Sigma-Aldrich). Cells of the non-adherent fraction (NAF), which contained T cells, were separated by washing and were stored at −80 °C in a freezing medium consisting of 11% human serum albumin (HSA; CSL Behring, King of Prussia, PA, USA), 20% DMSO (Sigma-Aldrich) and 10% glucose (Merck, Rahway, NJ, USA), and used for T cell stimulatory capacity experiments (see below). Monocyte-derived DCs were generated by culturing adherent monocytes for 4 days in the DC medium in the presence of a cytokine mixture of 800 IU/mL of recombinant human GM-CSF and 250 IU/mL of recombinant human IL-4 (day 0, both from Mitenyi Biotec, Bergisch Gladbach, Germany). On day 3, 400 IU/mL of fresh GM-CSF and 250 IU/mL of IL-4 were added to the cells. On day 4, iDCs were treated with 1 µM BP, 100 nM FICZ, or 500 µM I3S (BP and I3S were from Sigma-Aldrich, and FICZ was from InvivoGen, San Diego, CA, USA). The concentration of each compound was chosen after a careful consideration of the existing literature and a performance of preliminary studies to determine dosages with no adverse effects on cell viability [21,22,36,39]. Additionally, kinetic assays revealed a high induction of *AHRR* mRNA expression in both the I3S- and FICZ-treated DCs after 24 h. Thus, cells were only treated for a 24 h period to avoid a AhRR-mediated feedback inhibition of the investigated pathways. DMSO was used as vehicle control (mock) in all experiments (<0.2% [*v*/*v*]). To generate lipopolysaccharide-matured DCs (LPS-DCs), LPS was added (100 ng/mL; Sigma-Aldrich) after 6 h of primary treatment with different AhR ligands for an additional 16 h. Afterward, LPS-DCs were used to investigate phenotypical and functional modifications.

### 4.2. Flow Cytometric Analysis

Ligand-induced effects on DCs and on T cells following the DC–T cell co-culture were investigated by flow cytometry. To characterize phenotypic changes on the DCs, different panels including the following monoclonal antibodies (mAbs, all from BioLegend, unless otherwise stated) were used: CD11c-APC (3.9), CD11c-FITC (Bu15), HLA-DR-APC/Cy7 (L243), CD14-FITC (M5E2), CD80-PE (2D10), CD83-PE (HB15e), CD83-APC (HB15e), CD86-FITC (BU63), CD25-PE/Cy7 (BC96), CD39-PE (A1), CD73-PE (AD2), CCR7-PE/Cy7 (BD, 3D12), ILT3-APC (eBioscience, ZM4.1), ILT4-PE (42D1), and ILT5-PE (MKT5.1). Surface staining was performed in DPBS at 4 °C in the dark for 30 min, and the 7-AAD viability staining solution (Thermo Fisher Scientific, Waltham, MA, USA) was immediately added before analysis to identify dead cells. For the surface staining of T cells, the cells were incubated for 30 min at 4 °C in DPBS containing LIVE/DEAD Fixable Aqua Dead Cell Stain (Thermo Fisher Scientific) and following mAbs: CD3-FITC (UCHT1), CD4-APC/Fire (RPA-T4), CD25-PerCP (M-A251), CD69-PE/Cy7 (FN50), and CD45RO-PE (UCHL1). For intracellular staining, the cells were washed and treated with the eBioscience™ Foxp3/Transcription Factor staining buffer set according to the provided protocol. Subsequently, the cells were stained with FOXP3-APC (PCH101) mAb. The expression of the markers was directly determined using a FACSCanto II cell flow cytometer (BD Biosciences) and analyzed with the FlowJo software version 10.8.1 (FlowJo, LLC). The exemplified gating strategy and representative histogram overlays are depicted in Appendix A.

### 4.3. ATP Measurement

The ATP content in cell culture supernatants of AhR-ligand treated DCs was measured with CellTiter-Glo Luminescent Cell Viability Assay (Promega, Madison, WI, USA), with some modifications. After initial treatment with AhR ligands and stimulation with LPS, 2 × 10^5^ DCs were resuspended in the DC medium and seeded into 24-well plates. Then, 500 µM ATP was added to the cells, followed by incubation for 30 min at 37 °C, in 5% CO_2_. In the next step, the supernatant was collected and centrifuged at 4000 rpm, for 5 min, and subsequently, 100 µL of standards (0 to 10 µM) and the supernatant were transferred into a 96-well LumiNunc plate (Nunc, Thermo Fisher Scientific) in triplicates. Following this step, a 100 µL CellTiter-Glo reagent was added and mixed for 2 min on a rocker, and after incubating for 10 min at room temperature (RT), luminescence was measured using Infinite 200 PRO Microplate Reader (TECAN). The standard curve was used to determine the concentration of ATP in samples.

### 4.4. Transwell Migration Assay

To determine the effect of AhR activation on the migration capacity of DCs, we used a transwell migration assay as previously described [52]. Briefly, 100 µL of a migration medium (the DC medium supplemented with 400 IU/mL of GM-CSF, 250 IU/mL of IL-4, and 1% BSA (Roche Diagnostics, Basel, Switzerland)) was added to transwell inserts (Thermo Fisher Scientific) with a pore size of 5 µm. Then, the lower compartments were filled with 600 µL of the migration medium as well, and the wells were incubated for at least 30 min at 37 °C, in 5% CO_2_ to become warm and equilibrated. In the next step, a total of 2 × 10^5^ cells were transferred into insert wells in triplicates. To induce migration, 100 ng/mL of CCL19 (PeproTech, Cranbury, NJ, USA) was added to the lower wells, and the plates were incubated at 37 °C, in 5% CO_2_ for 2 h. Afterward, β-glucuronidase activity was used to quantify the migration of DCs. To this end, cells were harvested from the lower wells and lysed by the addition of 25 µL of DPBS and 5 µL of 1% Triton X-100 (Thermo Fisher Scientific). The lysate was spun down for 5 min, at 14,000 rpm, and supernatants were transferred to a 96-well plate (flat bottom). Then, 70 µL of *p*-nitrophenyl-β-D-glucuronide (Sigma-Aldrich) was added to each well and the plate was incubated for another 2 h at 37 °C, in 5% CO_2_. In the end, 100 µL of 0.4 M glycine at pH 10 (Carl Roth) was added to the wells to stop the enzymatic reaction, and absorbance was measured at 405 nm using Infinite 200 PRO Microplate Reader (TECAN). In each experiment, 2 × 10^5^ cells of each treatment were lysed in the same condition and used as positive controls.

### 4.5. sgRNA Sequences

Single-guide (sg) RNAs against human *AHR* were chosen according to the suggestions of the guide for RNA tools of Synthego. We used a multi-guide RNA strategy [53], to generate a larger fragment of deletion than that of standard indels, which led to an increase in efficiency. Both sgRNAs target exon 2 of *AHR* gene and their target sequences are 42 nucleotides apart from each other. The target sequences are GTAAAGCCAATCCCAGCTGA and TAATACAGAGTTGGACCGTT (replace T with U to get sgRNA sequences). SgRNAs were purchased from Sigma Aldrich.

### 4.6. CRISPR/Cas9-Mediated AHR Knock-Down

The knocking down of the *AHR* gene in DCs was performed by an adjusted experimental procedure from Royzman et al. [4]. Shortly, DCs were cultured as described before, and on day 4, cells were harvested from dishes and counted. For each nucleofection reaction, 1 × 10^6^ iDCs were resuspended in 20 µL of the primary nucleofection buffer P3 (with supplement added, Lonza). At the same time, Cas9 (Sigma Aldrich) and two sgRNAs were mixed to prepare Cas9 ribonucleoprotein particles (RNPs). RNP mixtures contained 1.2 μL of each sgRNA (120 pmol) and 5 μL recombinant Cas9 (26 pmol). Then, cells were mixed with RNPs and incubated at RT for 15 min. In the next step, cells were transferred into a 16-well-format nucleofection stripe for the Lonza 4D X unit (1 × 10^6^ iDCs each ligand treatment). Equally, for the control cells, DPBS was mixed with Cas9, and both conditions were nucleofected with the EA-100 code. After nucleofection, wells were rapidly filled up to 80 µL with prewarmed RPMI 1640 for 30 min at 37 °C. Subsequently, cells were pipetted into one well of a 24-well flat-bottomed culture plate having 420 μL of a pre-warmed DC medium and rested for one day at 37 °C, in 5% CO_2_ (four wells for CRISPRed cells and the same for control conditions). On the second day, all cells from *AHR*-KD and control conditions were harvested and then combined into one *AHR*-KD and one control pool, respectively, to assure equally distributed cells regarding CRISPR/Cas9 efficiency. Afterward, both *AHR*-KD and control cells were divided into four wells and treated with DMSO, 1 µM BP, 100 nM FICZ, and 500 µM I3S, and after 6 h, LPS was added to generate matured DCs. After 16 h post-LPS treatment, LPS-DCs were used for further experiments.

### 4.7. Sequencing of Edited DCs (Determination of Editing Efficiency)

Following CRISPR/Cas9 editing, part of the cells was lysed with the RLT-Plus buffer (Qiagen, Hilden, Germany), and Qiagen DNeasy Blood & Tissue Kit was used to extract the DNA. Next, using PCR and specific primer pairs, genomic regions neighboring the cut site were amplified. Primer pairs were designed by the Geneious software version 8 to amplify a 700- to 800-base-pair region guaranteeing that both sgRNAs induced cut sites would be included (forward: 5′-TCGGAAGAATTTAACCCATTCCC-3′; reverse: 5′-CAGAAAATCCAGCAAGATGGTGT-3′). Subsequently, the PCR product was purified using PCR Purification Kit (Qiagen), and 75 ng DNA was sent to Eurofins Scientific for sequencing. Next, the efficiency of the gene editing was calculated by using the Synthego online tool *Inference of CRISPR Edits (ICE)* (V3.0).

### 4.8. Quantitative Real-Time PCR (qPCR)

The ligand-treated control and *AHR*-KD DCs were lysed with the RLTPlus buffer (Qiagen), and total RNA was isolated by using RNeasy Plus Mini Kit (Qiagen) according to the manufacturer’s protocol. A total of 1 µg of RNA was used for reverse transcription into cDNA using First Strand cDNA Synthesis Kit (Thermo Fisher Scientific). Subsequently, gene expression was determined by using 2x qPCR S’Green BlueMix (Biozym Scientific, Hessisch Oldendorf, Germany) on CFX96 Real-Time PCR Detection System (Bio-Rad). All primers were tested for specificity and quality according to the MIQE guidelines [54] (Table 1). Samples’ mRNA levels were normalized to the expression of the *CHCHD2* reference gene and/or to the mock control.

### 4.9. Human Mixed Lymphocyte Reaction (MLR)

Different ratios of WT and *AHR*-KD LPS-DCs pre-treated with 1 µM BP, 100 nM FICZ, 500 µM I3S, and DMSO were co-cultured with 4 × 10^5^ allogenic NAF-derived cells (with each condition triplicated), in 96-well cell culture plates (Thermo Fisher Scientific) for 72 h, at 37 °C, in 5% CO_2_. Dynabeads Human T-Activator CD3/CD28 (Thermo Fisher Scientific) was added to the T cells to induce proliferation as a positive control. To assess the direct effect of the AhR ligands on T cell proliferation, each ligand was added to Dynabead-activated T cells as ligand controls. After 72 h of culture, proliferation was measured using the Cell Proliferation ELISA BrdU kit (colorimetric, Roche) in accordance with the manufacturer’s instructions. Proliferation was measured at 450 nm using 570 nm as a reference wavelength (TECAN).

### 4.10. DC–T Cell Co-Cultures

To investigate the potential of AhR ligand-treated DCs in regulatory T cell induction, dendritic cells were co-cultured with T cells. Using Human CD4^+^ T cell Isolation Kit, in accordance with the provided protocol (Miltenyi Biotec, Bergisch Gladbach, Germany), CD4^+^ T cells were isolated from NAF. Afterward, 1 × 10^6^ CD4^+^ T cells were co-cultured in a 24-well plate at a ratio of 10:1 with LPS-matured DCs pre-treated with 1 µM BP, 100 nM FICZ, 500 µM I3S, and DMSO. DMSO-treated matured DCs that were co-cultured with CD4^+^ T cells in the presence of each ligand were used as controls to exclude any direct effect of each ligand on regulatory T cell induction. After five days of co-culture, supernatants were isolated and stored at −80 °C for further cytokine measurement. Then, cells were harvested, stained with mAbs, and analyzed by flow cytometry.

### 4.11. Western Blot

Following iDC generation, LPS was added for 16 h to the cells which were primarily treated with DMSO (mock), 1 µM BP, 100 nM FICZ, and 500 µM I3S. Then, the equivalent cell numbers (1.5 × 10^6^) of WT cells and also (1 × 10^6^) *AHR*-KD and control cells (knocking down *AHR* by the CRISPR/Cas9 strategy) pre-treated only with DMSO (mock) and I3S, and matured with LPS (100 ng/mL), were used for total protein extraction. For this, cells were lysed, at least for 1 h at 4 °C in a sodium deoxycholate lysis buffer, consisting of PBS, 150 mM NaCl, 50 mM TRIS (pH 8.0), 1% Triton X-100, 0.1% SDS, 20 mM sodium orthovanadate, 2 mM phenylmethylsulfonyl fluoride, 2 mM sodium fluoride, 12.5 U/mL of Benzonase, and 1 mM MgCl2. Then, the protein concentration of the lysate was determined using Pierce BCA Protein Assay Kit (Thermo Fisher). Following this step, lysates were denatured in a reducing Laemmli buffer at 95 °C for 5 min. Afterward, proteins were separated via SDS-polyacrylamide gel electrophoresis and blotted onto a nitrocellulose membrane (GE Healthcare). As a blocking reagent, the membranes were blocked with ROTI Block for 1 h and then were probed overnight at 4 °C with the subsequent antibodies: mouse anti-human/mouse AhR antibody (Santa Cruz, cat#sc-133088), and mouse anti-mouse/human GAPDH (Millipore, cat#MAB374). Later, the membranes were washed, and the appropriate HRP-labeled secondary antibody (horse anti-mouse IgG (Cell Signaling, cat#7076)) was added. Specific signals were detected using ECL Prime Western Blotting Detection Reagent (GE Healthcare). Fully uncropped Western blot images are shown in Appendix A.

### 4.12. Statistics

Statistical analyses were performed and presented using GraphPad Prism 9.3.1. Data are expressed as mean values ± SEM. According to the data distribution, appropriate statistic tests (*t*-test/Mann–Whitney test or 1-way ANOVA/Kruskal–Wallis test) were used. A value of *p* < 0.05 was considered statistically significant.

## Figures and Tables

**Figure 1 ijms-24-07801-f001:**
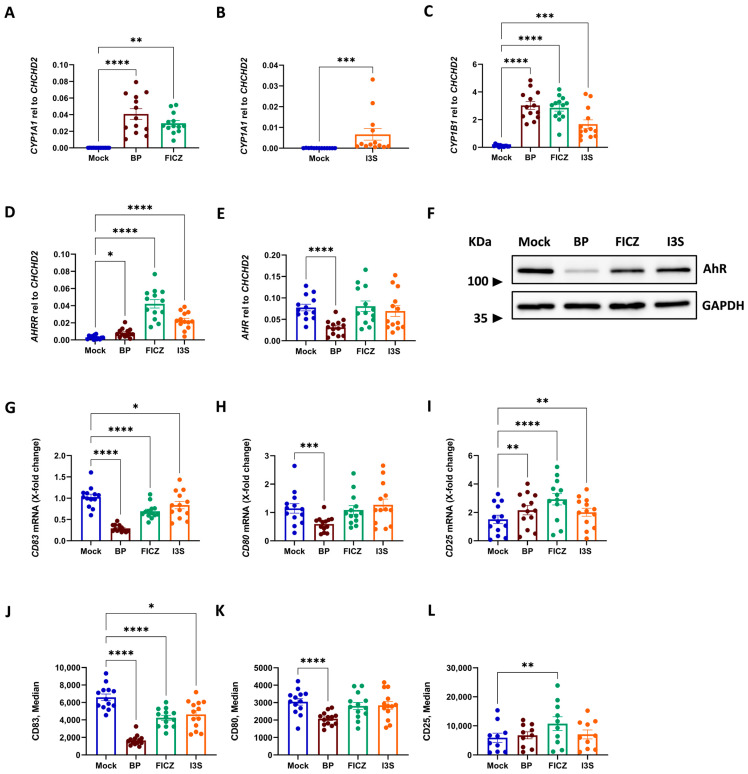
AhR-induced phenotypic changes in DCs. DCs were exposed to AhR ligands (1 µM BP, 100 nM FICZ, and 500 µM I3S), and DMSO (vehicle control, mock) for 6 h which was followed by 16 h of LPS (100 ng/mL) treatment. (**A**–**D**) qPCR analysis of AhR-induced transcripts including *CYP1A1*, *CYP1B1*, and *AHRR*, performed and normalized to relative levels of reference gene *CHCHD2*. (**E**) Expression of *AHR* mRNA normalized to relative levels of reference gene *CHCHD2*. (**F**) Protein expression of AhR investigated using Western blot analyses. LPS-matured DCs (1.5 × 10^6^) pre-treated with different ligands or DMSO (mock) harvested 24 h post-ligand treatment. (**G**–**I**) qPCR analysis of DCs’ surface markers including *CD83*, *CD80*, and *CD25* (X-Fold change). (**J**–**L**) Flow cytometric analyses of CD83, CD80, and CD25 (median gated on CD11c^+^ cells). The relative mRNA expression of the mock control was set to one. Data are represented as mean ± SEM. * *p* < 0.05, ** *p* < 0.01, *** *p* < 0.001, and **** *p* < 0.0001; bars without annotation are not significant (*p* > 0.05). Statistical significance was tested using the *t*-test/Mann–Whitney test or a 1-way ANOVA/Kruskal–Wallis test.

**Figure 2 ijms-24-07801-f002:**
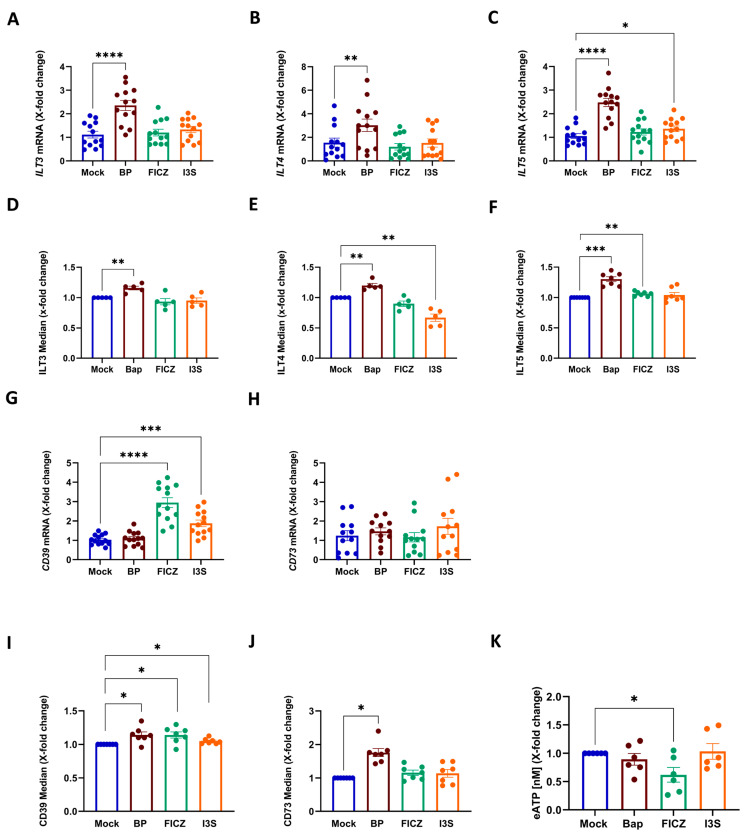
BP induced the strongest tolerogenic phenotype in DCs. (**A**–**C**,**G**,**H**) Analysis of ligand-treated LPS-matured DCs regarding the expression of negative regulators including ILT3 (CD85k), ILT4 (CD85d), ILT5 (CD85a), CD39, and CD73, both on the mRNA (determined via qPCR, normalized to the relative levels of the mock) and (**D**–**F**,**I**,**J**) protein levels (determined via flow cytometry, with the median normalized to the relative levels of mock, and the median gated on CD11c^+^ cells). (**K**) Investigation of cells analyzed in (**A**–**J**) regarding their ATP content. Data were first normalized to an ATP standard curve ranging from 0 to 10 µM and then normalized to the relative levels of the mock. The relative mRNA expression of the mock control was set to one. Data are represented as mean ± SEM. * *p* < 0.05, ** *p* < 0.01, *** *p* < 0.001, and **** *p* < 0.0001; bars without annotation are not significant (*p* > 0.05). Statistical significance was tested using the Mann–Whitney test or 1-way ANOVA. ATP; adenosine triphosphate.

**Figure 3 ijms-24-07801-f003:**
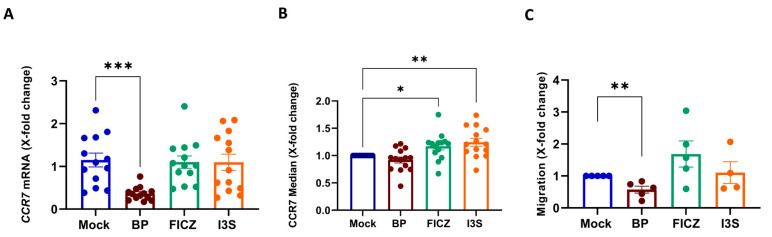
BP-treated cells revealed reduced migratory capacity. DCs treated for 6 h with 1 μM BP, 100 nM FICZ, 500 µM I3S, or DMSO (mock), 16 h post-LPS treatment (100 ng/mL), were harvested and used to assess (**A**) *CCR7* expression on mRNA levels and (**B**) protein levels. (**C**) Quantification of migration using a β-glucuronidase assay. Data are represented as mean ± SEM. * *p* < 0.05, ** *p* < 0.01, and *** *p* < 0.001; bars without annotation are not significant (*p* > 0.05). Statistical significance was tested using a Mann–Whitney test or 1-way ANOVA.

**Figure 4 ijms-24-07801-f004:**
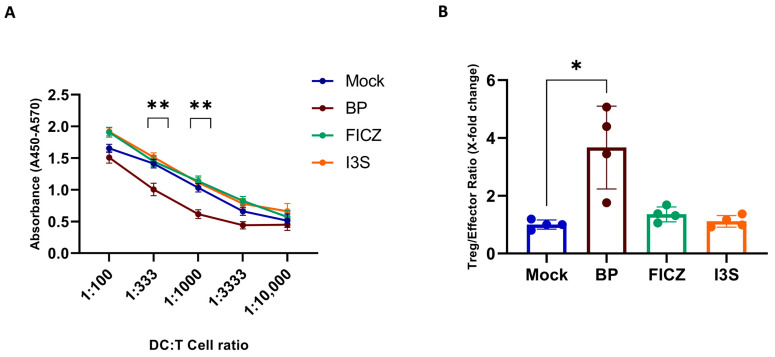
BP-treated DCs alter T cell proliferation and differentiation. Human iDCs were pre-treated with 1 μM BP, 100 nM FICZ, or 500 μM I3S or DMSO (mock) for 6 h before LPS (100 ng/mL) was added for a further 16 h. DCs were co-cultured with allogeneic NAF cells (4 × 10^5^) at different ratios (1:100 to 1:10,000) for 72 h. (**A**) Assessment of NAF-derived cell proliferation using the BrdU proliferation assay. (**B**) Co-culture of LPS-matured DCs pre-treated with different AhR ligands or DMSO (mock) with MACS-isolated allogeneic CD4^+^ T cells for 120 h, followed by flow cytometric analyses. The ratio of Treg (CD25^high^FoxP3^+^) to effector T cells (CD45RO^+^CD69^+^) is shown. Data are represented as mean ± SEM. * *p* < 0.05 and ** *p* < 0.01; bars without annotation are not significant (*p* > 0.05). Statistical significance was tested using a *t*-test or 1-way ANOVA.

**Figure 5 ijms-24-07801-f005:**
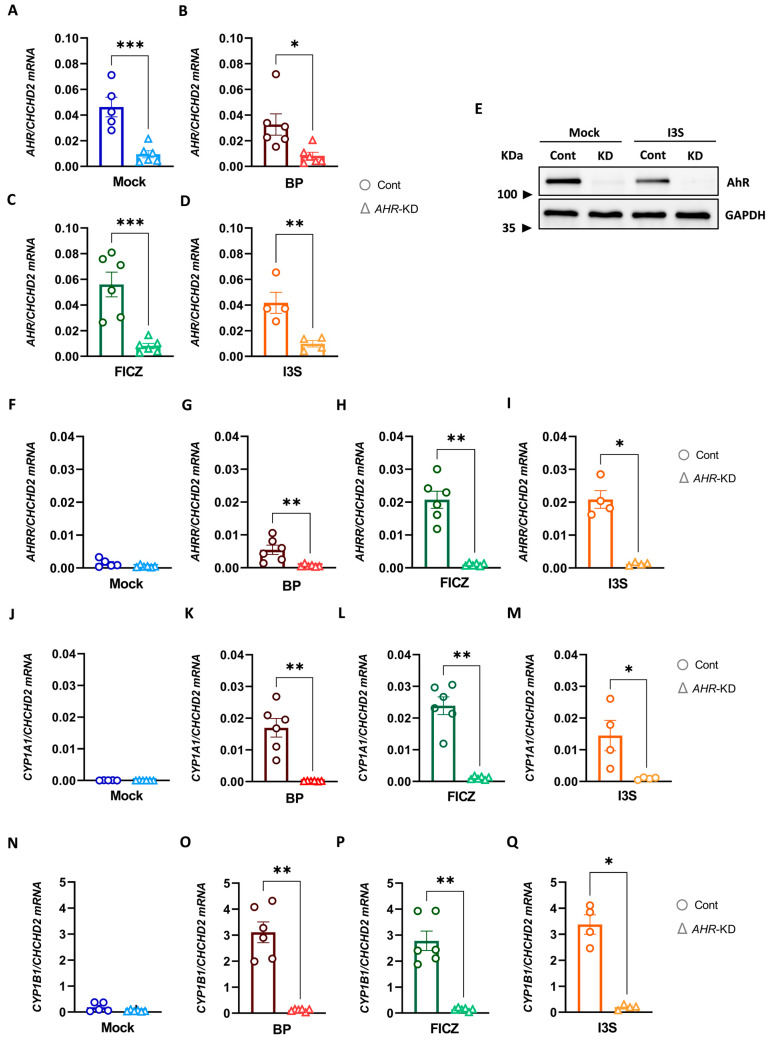
Effects of AhR-specific knock-down in hMoDCs. iDCs were generated from PBMCs in the presence of GM-CSF and IL-4, then *AHR* was knocked down using CRISPR/Cas9 strategy. Afterward, iDCs were treated with 1 μM BP, 100 nM FICZ, and 500 μM or DMSO (mock) for 6 h, followed by 16 h of LPS exposure (100 ng/mL). (**A**–**D**) Evaluation of mRNA expression of *AHR* in both control (Cont) and *AHR*-KD cells via qPCR. (**E**) Assessment of knock-down efficiency on protein level via harvest and lysis of cells 24 h after treatment (I3S or mock) to determine AhR expression. (**F**–**Q**) mRNA expression of AhR-induced transcripts (*AHRR*, *CYP1A1*, and *CYP1B1*) by *AHR*-KD cells or control cells. Data are represented as mean ± SEM. * *p* < 0.05, ** *p* < 0.01, and *** *p* < 0.001. Statistical significance was tested using a *t*-test.

**Figure 6 ijms-24-07801-f006:**
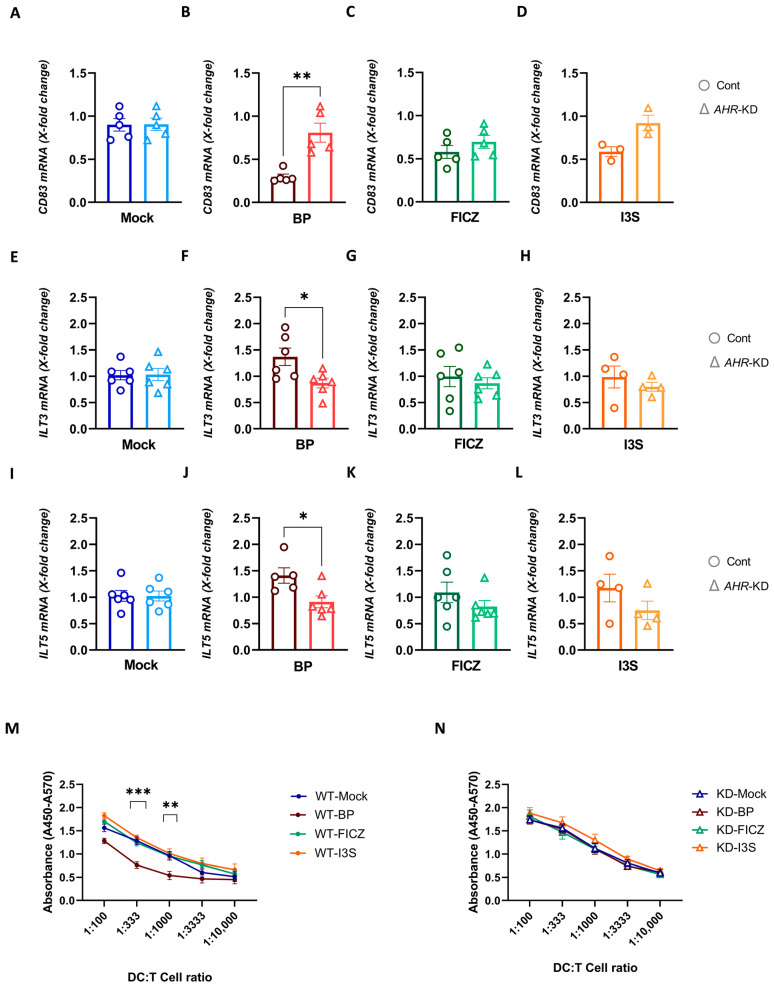
Deletion of *AHR* in DCs reverses the tolerogenic effects of BP. *AHR* was knocked down in iDCs using CRISPR/Cas9 strategy. Afterward, iDCs were treated with, 1 μM BP, 100 nM FICZ, and 500 μM I3S or DMSO (mock) for 6 h, followed by 16 h LPS exposure (100 ng/mL). (**A**–**L**) qPCR evaluation of mRNA expression levels of *CD83*, *ILT3*, and *ILT5*, in both control and *AHR*-KD cells. (**M**,**N**) Co-culture of ligand-treated LPS-matured DCs with 4 × 10^5^ allogeneic NAF-derived cells for 72 h and assessment of T cell proliferation using the BrdU cell proliferation ELISA kit. Data are represented as mean ± SEM. * *p* < 0.05, ** *p* < 0.01, and *** *p* < 0.001. Statistical significance was tested using a *t*-test or 1-way ANOVA.

**Figure 7 ijms-24-07801-f007:**
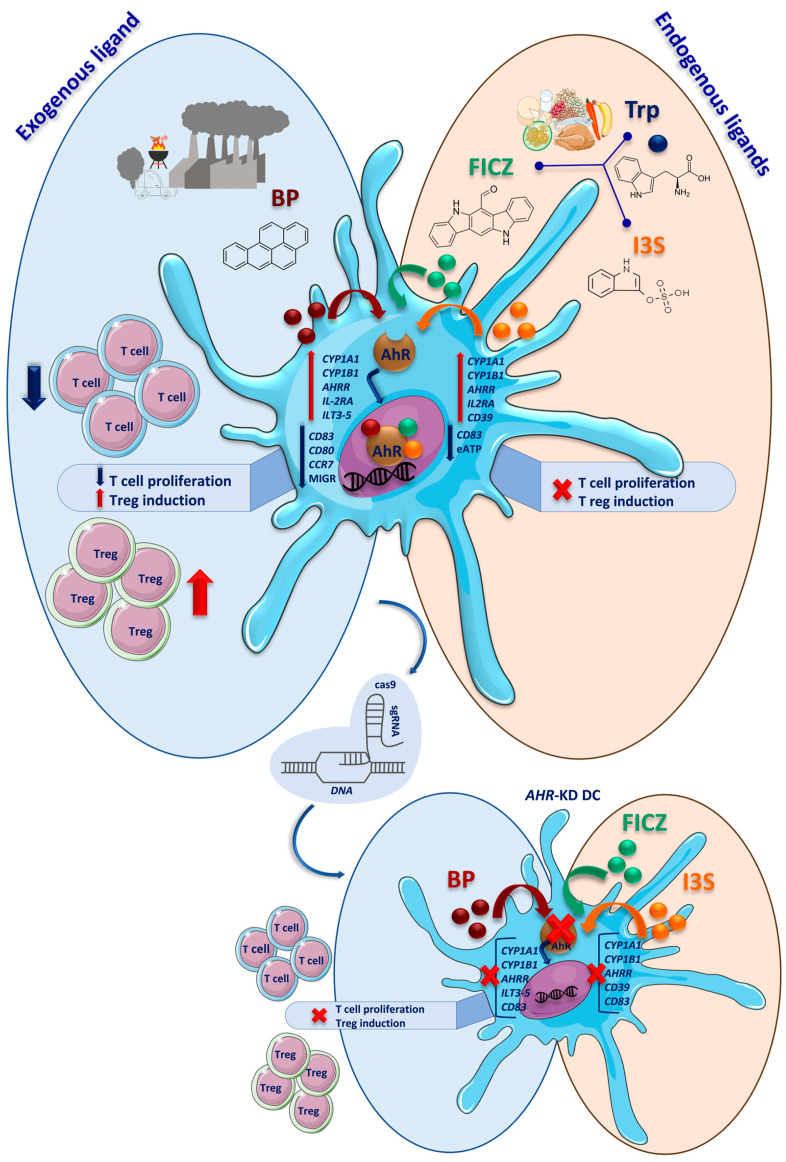
Graphical overview. Agonists of AhR can be classified based on their source into two groups: exogenous agonists (natural or synthetic), and endogenous agonists (host or microbiota-derived). Treating DCs with three different AhR ligands (BP, FICZ, and I3S) activates AhR in these cells, which is shown by the expression of AhR-induced transcripts (*CYP1A1*, *CYP1B1*, and *AHRR*). Following AhR activation, regulatory changes in DCs’ phenotypic markers (downregulation of *CD83* and upregulation of *CD25*) qwere detected. These alterations were accompanied by the induction of negative regulators (*ILTs* and ectonucleotidases), especially in BP-treated cells. An observation of restricted T cell proliferation and an increase in the Treg to effector T cell ratio in DC–T cell co-cultures indicated a tolerogenic phenotype in these DCs which is more prominent in BP-treated ones. Using the CRISPR/Cas9 strategy, we ruled out the possibility of inhibitory effects of each ligand itself (especially BP), and once more emphasized the regulatory effect of AhR activation in DCs. Transcriptional changes are highlighted by italicization. Reduced migration capacity (MIGR) was seen only in BP-treated DCs. Reduction in the eATP level only was observed significantly in FICZ-treated DCs but not I3S-treated ones.

**Table 1 ijms-24-07801-t001:** Primer sequences for qPCR.

Gene.	Orientation	Sequence
** *AHR* **	Forward	5′-ACATCACCTACGCCAGTCG-3′
	Reverse	5′-CGCTTGGAAGGATTTGACTTGA-3′
** *AHRR* **	Forward	5′-GCGCCTCAGTGTCAGTTACC-3′
	Reverse	5′-GAAGCCCAGATAGTCCACGAT-3′
** *CYP1A1* **	Forward	5′-TCATCCCCTATTCTTCGCTACC-3′
	Reverse	5′-TCTCCTGACAGTGCTCAATC-3′
** *CYP1B1* **	Forward	5′-AACGTACCGGCCACTATCAC-3′
	Reverse	5′-TCACCCATACAAGGCAGACG-3′
** *CD83* **	Forward	5′-TGCTGCTGGCTCTGGTTATT-3′
	Reverse	5′-TGTGAGGAGTCACTAGCCCT-3′
** *CD80* **	Forward	5′-CCATCCAAGTGTCCATACCTC-3′
	Reverse	5′-GCCAGCTCTTCAACAGAAAC-3′
** *CD25 (IL2RA)* **	Forward	5′-ACTTCCTGCCTCGTCAC-3′
	Reverse	5′-TCTACTCTTCCTCTGTCTCCG-3′
** *CCR7* **	Forward	5′-GCTCTCCTTGTCATTTTCCAG-3′
	Reverse	5′-GCTTTAAAGTTCCGCACGTC-3′
** *CD39* **	Forward	5′-AGAGGAAGGTGCCTATGGCT-3′
	Reverse	5′-TGGGGACTCGATAGTCTGGTT-3′
** *CD73* **	Forward	5′-AGTACCAGGGCACTATCTGGT-3′
	Reverse	5′-TGAGGAGTGGCTCGATCAGT-3′
** *ILT3* **	Forward	5′-CTTCAGCTCACACGGCTTCT-3′
	Reverse	5′-ACTGACCCTGTAGGCATGAG-3′
** *ILT4* **	Forward	5′-CTCAACTCCGACCCCTACCT-3′
	Reverse	5′-AAGATGCCGATCACAACCCC-3′
** *ILT5* **	Forward	5′-GACAGAGCCCAAGGACAG-3′
	Reverse	5′-GGGTCTTCATCGTGTGGG-3′
** *CHCHD2* **	Forward	5′-CACATTGGGTCACGCCATTA-3′
	Reverse	5′-GCTTGATGTCACCCTGGTTCT-3′

AHR: aryl hydrocarbon receptor; AHRR: aryl hydrocarbon receptor repressor; CYP1: cytochrome P450 enzymes family 1; ILT: immunoglobulin-like transcript (ILT3 (CD85k), ILT4 (CD85d), and ILT5 (CD85a)); CHCHD2: coiled-coil-helix-coiled-coil-helix domain containing 2.

## Data Availability

All data have been presented within the manuscript.

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
