# Peer review of "Differential Modulation of Dendritic Cell Biology by Endogenous and Exogenous Aryl Hydrocarbon Receptor Ligands"

_ijms, 2023, doi:10.3390/ijms24097801_

Round 1

Reviewer 1 Report

In this manuscript, the authors described the impact of endogenous and exogenous AhR ligands on the phenotype and the function of LPS-treated MoDCs. They concluded that AhR ligands impact differently on the maturation program induced by LPS. Overall, the paper is clearly written.

Here are some concerns :

- Concerning the impact of AhR ligands on the DC phenotypes, I strongly encourage authors to show representative phenotypes of DCs exposed to different AhR ligands, in addition to statistical diagrams. Same suggestion as before with ILT3, 4, 5 markers and CCR7.

- Concerning the MLRs results, the authors describe an impact of DC exposed to different AhR ligands on the proliferation of T cells. The authors should also show the impact of treated DCs on T cell polarization, by measuring the Th1, Th2, Treg and Th17 cytokines in the co-culture supernatants.

- The authors used only one source of DC maturation (LPS). This is a major limitation of the study, since it is quite conceivable that these AhR ligands have a different effect depending on the differentiation program triggered in the DCs. This concern should appear in the Discussion section. It also seems important to make it appear in the Abstract that the study was carried out on human MoDCs and that the impact of AhR ligands was evaluated on LPS-treated DCs.

Author Response

Point1: Concerning the impact of AhR ligands on the DC phenotypes, I strongly encourage authors to show representative phenotypes of DCs exposed to different AhR ligands, in addition to statistical diagrams. Same suggestion as before with ILT3, 4, 5 markers and CCR7.

Response1: We would like to thank to the reviewer for providing this valuable suggestion, and we have now included the gating strategy along with representative histogram graphs in the supplementary figures (Figure S4-6). 

Point2: Concerning the MLRs results, the authors describe an impact of DC exposed to different AhR ligands on the proliferation of T cells. The authors should also show the impact of treated DCs on T cell polarization, by measuring the Th1, Th2, Treg and Th17 cytokines in the co-culture supernatants.

Response 2: This is a valid point. However, when we measured cytokines from the respective supernatant using a cytometric bead array (CBA, LEGENDplex HU Th Cytokine Panel), was revealed no differences between mock cells and ligand-treated cells. Therefore, we suggest that AhR-ligands impair T cell stimulatory capacity rather than T cell differentiation.

Point3: The authors used only one source of DC maturation (LPS). This is a major limitation of the study, since it is quite conceivable that these AhR ligands have a different effect depending on the differentiation program triggered in the DCs. This concern should appear in the Discussion section. It also seems important to make it appear in the Abstract that the study was carried out on human MoDCs and that the impact of AhR ligands was evaluated on LPS-treated DCs.

Response 3: We would like to extend our sincere gratitude to the reviewer for providing this important comment. These points have been addressed in the manuscript.

Reviewer 2 Report

This study addressed the effect of three AHR ligands, BP, FICZ, and I3S on DC responses using human monocyte derived DCs. The manuscript is well written. It is interesting that only BP increased the ratio of Treg in co-culture system of DC and T cells. However, this reviewer concerns several points related with data presentation.

(1) In FACS data, representative histogram and/or dot blot data for each sample-stimulated DC, and gating strategy should be shown in supplemental files. Does “median gated on living cells” indicate “ mean fluorescence intensity”? Expression changes in some molecules (e.g., CD25, ILT and CCR7) look low. 

(2) In western blotting, were GAPDH and Ahr detected in the same membrane? In supplemental data, molecular size markers should be included in each membrane. Fig. 5E, 105 and 35 indicate molecular size?

(3) How were the concentrations of BP, FICZ, and I3S set up?

(4) FACS analysis was not performed in AHR-depleted DC?

(5) Fig.7. The results of gene expression and protein expression are mixed up. The authors should describe it fairly and should not pick up positive data only. The effects of FICZ, and I3S are different, and it should also be reflected in the figure.

(6) Did the authors perform a co-culture experiment of DCs and syngeneic T cells?

Round 2

Reviewer 1 Report

The authors have satisfactorily addressed all my concerns

Reviewer 2 Report

The authors replied to comments by this reviewer and improved the quality of manuscript.